# Predominance of Calcium Pyrophosphate Crystals in Synovial Fluid Samples of Patients at a Large Tertiary Center

**DOI:** 10.3390/diagnostics15070907

**Published:** 2025-04-01

**Authors:** Tobias Manigold, Alexander Leichtle

**Affiliations:** 1Department of Rheumatology and Immunology, Inselspital, Bern University Hospital, University of Bern, 3008 Bern, Switzerland; 2Department of Clinical Chemistry, Inselspital University Hospital, University of Bern, 3008 Bern, Switzerland; 3Laboratory Medicine, Cantonal Hospital Baden (KSB.ch), 5404 Baden, Switzerland; 4Center for Artificial Intelligence in Medicine, University of Bern, 3008 Bern, Switzerland

**Keywords:** crystal arthropathies, gout, pseudogout, CPPD, synovial analysis

## Abstract

**Background:** Crystal arthritides represent the most common inflammatory rheumatologic condition. While the prevalence of gouty arthritis by monosodium urate (MSU) is well established, the prevalences of calciumpyrophosphat (CPP) and basic calcium pyrophosphate (ARP) arthritis are less clear. We herein sought to assess the prevalence and inflammatory characteristics of crystal arthritides at our institution, the biggest tertiary center in Switzerland. **Methods:** A total of 5036 synovial fluid (SF) samples were analyzed with regard to crystal positivity as well as joint, age, and sex distribution in affected patients. We furthermore compared inflammatory and non-inflammatory SF samples for yields of their Polymorphonuclear (PMN) fractions. **Results:** About half of all samples were derived from knee joints, a male/female ratio up to 10.1:1 among the MSU-positive, and a clear shift towards elder patients with CPP–arthritis was seen. These findings were in line with previous studies and suggest good comparability of our cohort. Of note, 21.9% of all samples were CPP positive, whereas 15.3% and 9.5% were positive for MSU and ARP/alizarin-red positive, respectively. Importantly, CPP crystals were predominant in inflammatory (58.9%) and non-inflammatory (65.7%) samples. By contrast, MSU crystals were significantly more often associated with synovitis (*p* < 0.001). Interestingly, higher PMN fractions were found in non-inflammatory MSU-positive samples (*p* < 0.01), whereas a similar trend was seen in CPP-positive samples. **Conclusions:** CPP arthritis represented the most frequent crystal arthritis form at our center. Higher PMN fractions in non-inflammatory samples with CPP and MSU crystals suggest subclinical inflammation and provide further arguments for earlier anti-inflammatory and uric acid-lowering therapies in patients with crystal deposits.

## 1. Introduction

Crystal arthritides by monosodium urate (MSU arthritis, “gout”) or calciumpyrophosphate (CPP arthritis, “pseudogout”) crystal deposits represent the most prevalent causes of arthritis. Specifically, gouty arthritis is the most common arthritis form, with an estimated prevalence of more than 5% in the US [1]. In contrast, the prevalence of CPP arthritis is not really known, whereas the prevalence of chondrocalcinosis (CC), a pre-state of asymptomatic deposition of calcium-containing crystals in cartilage structures, reaches up to 7% [2]. Importantly, conventional radiographic (CR) studies indicate a prevalence of 15% in persons around 60 years of age and increases by 15–20% per subsequent decade [3]. Axial affection was described to be as high as 60% in persons > 70 years of age [4]. Of note, an OMERACT study showed that ultrasound is more sensitive than CR for the detection of CC [5], indicating that the prevalence of CC may even be underestimated by earlier studies using CR.

Basic Calcium Phosphate (ARP) is a normal component of bones and teeth matrix. Consequently, ARP is frequently found in synovial fluid of osteoarthritic joints without causing synovitis in most cases. However, ARP can cause periarthritis around hip joints and possible inflammatory symptoms with large effusions in shoulder joints, known as Milwaukee shoulder syndrome. Likewise, CPP crystal deposits are often found in the menisci of the knee. With increasing osteoarthritis, crystals may be eluted from the menisci into the joint space and then found in synovial fluid without causing synovitis.

Detection of crystals in synovial fluid (SF) represents the gold standard and basis for defining crystal arthritides and associated pathologies. Although dual energy-CT (DECT) and ultrasound detection of double contour signs are recognized as alternative modalities to diagnose deposition of MSU crystals, these technologies are increasingly capable of distinguishing CPP from other calcium-containing crystal types, such as ARP [6,7]. Crystal analysis in SF is based on crystal shape and birefringence pattern under compensated polarized microscopy (for MSU, CPP) or alizarin-red staining (of ARP) under ordinary or polarized light microscopy [8]. Combination with leucocyte counts and microbiological cultures allows the diagnosis of a crystal arthritis form and distinction from septic arthritis. However, negative results for crystal analysis can be explained by sampling errors, high inter-operator variability, low crystal load, and poor birefringence—especially by CPP crystals [9]. Daily rheumatology practice and few studies suggest that CPP arthritis is probably under-recognized. Furthermore, a high percentage of overlapping or mimicking CPP arthritis is suggested by recent reports of higher yields of CC in patients with negative rheumatoid arthritis [10,11]. Likewise, higher percentages of MSU crystals are found among patients with psoriatic arthritis (PsA) [12]. Thus, crystal arthritides can lead to confounded activity assessments of other inflammatory diseases and to inadequate treatment decisions. In addition, an increasing body of evidence suggests that inflammation by both gout and pseudogout leads to a plethora of cardiovascular and other morbidities as well as increased mortality [13,14,15,16].

Taken together, crystal arthritides have a significant clinical impact and are likely to become more important in the future. Basic knowledge of the prevalence of these diseases and the affected patients may help to better identify patients at risk and, therefore, contribute to earlier diagnosis and treatment. We herein retrospectively analyzed data from our institution, the biggest tertiary center in Switzerland and one of the biggest SF cohorts reported so far. We aimed for a clinically unbiased analysis with regard to sex, age, diagnosis, symptomsand indication for arthrocentesis in order to avoid possible selection bias with effects on the results.

## 2. Methods

### 2.1. Synovial Fluid Samples

We obtained the data from the Insel Data Science Center (IDSC) after regulatory checks and approval by the local ethics committee (No. 2023-01815). Specifically, we retrospectively analyzed all arthrocentesis samples between January 2015 and August 2023 at the central laboratory of the Inselspital Bern, for which crystal analysis and cellular analyses were available.

In total, 5479 synovial fluid samples from 3579 patients obtained throughout eight years (2015–2023) were studied. Samples were accepted and collected via arthrocentesis from all departments across the Inselspital University Hospital, Bern, Switzerland, as well as external admissions in the surroundings. Samples that rejected or revoked general informed consent were excluded from subsequent analyses.

### 2.2. Crystal Detection

Crystal analysis was performed by rheumatologists and specifically trained lab personnel. While assessing rheumatologists was changing over time, lab personnel (3–4 lab staff members) responsible for most of the assessments was mostly stable throughout the observation time. Internal studies revealed good comparability of results among staff members.

All samples for which crystal assessment was requested were applied to centrifugation using a Centry 103 Microcentrifuge (Gilson, Middleton, WI, USA) at 6000 rpm for 1 min. Subsequently, supernatants were removed, and cell pellets were applied to a cover slip and applied to a rotator stage of a Leica DM/LSP (Leica Microsystems, Wetzlar, Germany) microscope equipped with rotating stage, a lambda polarizer, and a compensator, allowing for the detection of negative and positive birefringence as well as extracellular and intracellular crystal location under 400× magnification. Crystals were assessed based on shape and negative (MSU) or positive (CPP) birefringence under compensated polarized light microscopy (CPLM).

Alizarin-red staining was only performed when specifically requested by the ordering physician. Specifically, before staining, alizarin-red solution of 1% was filtered (Millipore, Burlington, MA, USA, Millex-GP, 0.22 μm) in order to eliminate possible precipitates. Detection of ARP/alizarin-red positive crystals was performed under ordinary light microscopy. Of note, alizarin-red staining is not specific for ARP crystals but also includes other calcium-containing crystals, including CPP. Although ARP crystals were, based on size and shape, mostly considered basic-calcium phosphate (BCP) crystals, we used the term ARP instead of BCP. Samples were considered ARP when roundish, dark alizarin-red staining deposits were detected by plain light microscopy. Importantly, throughout the entire study, only 13 CPP/ARP double positive and 58 alizarin-positive/CPP-negative samples were detected, suggesting a low false positive rate. All results were documented in a semiquantitative manner (little, moderate, massive) at the discretion of the observer.

### 2.3. Leucocyte Analysis

In parallel, leucocyte counts, including fractions of mononuclear and Polymorphonuclear cells (PMN), were assessed using an automated cell counter (Sysmex, Norderstedt, Germany) in most SF samples. Analysis was performed independent of the indication of arthrocentesis. Should read: Subsequently, crystal-positive, inflammatory (>2000 cells/μL) and non-inflammatory (<2000 cells/μL) samples, were compared for male/female and age distributions, the prevalence of the different crystal types, as well as PMN fractions and localization (intracellular or intra-/extracellular or extracellular) of crystals as reported by the observers. The sample analysis algorithm is depicted in Appendix A.

### 2.4. Statistical Analyses:

We used R (v.4.4.1) for our statistical analyses. The Sankey diagram (Appendix A) was generated using the package networkD3. Statistical analyses included the chi-square test and the Kruskal–Wallis test, as indicated.

## 3. Results

A total of 5579 samples was screened, of which 5036 positive informed consent was available and, thus, could be enrolled into the study (Appendix A) and derived from 3674 different patients, of which 58.8% were male (mean 63.9, range 18–102 years) and 41.2% were female (mean 65.1, range 18–97 years).

Based on non-systematic, non-mandatory free-text entries of the requesting physician, information on the anatomical site of arthrocentesis was available in 4295/5036 (85.2%) of samples. As expected, knee joints contributed the majority of all samples (48.7%), followed by “other”. Of note, in our clinical information system, the documentation of the anatomical site of arthrocentesis was non-systematic and based on free-text information. This information, therefore, included rare structures, imprecise descriptions, misspelled terms, or no information at all. In these cases, we defined the anatomical source of SF as “other”.

In 3623/5036 (71.9%) analysis for MSU and CPP crystals was requested by the ordering physician, while ARPARP analysis was requested for 791/5036 (15.7%) samples. Interestingly, among 3623 samples, 794 (21.9%) and 555 (15.3%) were positive for CPP for MSU crystals, respectively (Figure 1a). Only 75 (9.5%) of 791 alizarin-stained samples were positive and, therefore, considered ARP positive.

Taken together and excluding possible double or triple crystal positive samples, a total of 1424/3623 samples contained crystals, resulting in a crystal prevalence of 39.3% in our cohort. Overall, 2611/5036 samples were analyzed for cellular composition, of which 1530 (58.6%; Figure 1b) were defined as inflammatory and 41.4% as non-inflammatory. The male/female ratios were similar among inflammatory and non-inflammatory samples.

### 3.1. Prevalence of Crystals According to Affected Joint

We next analyzed the distribution of crystal types according to the originating joint (Table 1). Knee and other joints contributed the majority of MSU and CPP crystals, including multiple positive crystal types (e.g., MSU/CPP).

Some of the prevalences seemed to coincide with clinical expectations. Namely, MSU and CPP crystals were predominant in wrist joints, whereas MSU crystals were more frequent in ankle, MTP I, and finger/toe joints. Likewise, albeit in very low numbers, ARP crystals were mostly found in the shoulder joint (Table 1) after the knee joint. Overall, CPP was the most predominant crystal type (52.9%), followed by MSU (35.7%). In addition, several multiple positive constellations were detected (Figure 1, Table 2), with CPP/MSU being the most prevalent (6.0%). Thus, 58.9–65.7% of samples contained CPP crystals, at least to some extent.

### 3.2. Prevalence of Crystal Arthritides

In order to analyze the prevalence of arthritides in the presence of crystals, we next compared inflammatory (i.e., with ≥2000 cells/μL) and non-inflammatory among crystal-positive samples. Importantly, crystal-negative samples were also examined for the presence of crystals but yielded negative results. Thus, we generally assumed that crystal-negative, non-inflammatory samples were vastly due to degenerative reasons (e.g., osteoarthritis), whereas crystal-negative, inflammatory samples vastly represented other forms of arthritides (e.g., rheumatoid or spondyloarthritis). As can be seen in Table 2, 58.9% of inflammatory samples contained CPP to some extent. Similar prevalences for CPP were found in the 201 non-inflammatory samples, as differences were non-significant. By contrast, MSU-positive samples were significantly (*p* < 0.001) more frequent among inflammatory than non-inflammatory samples. Of note, similar frequencies of CPP/ARP double-positive samples were found in both groups, consistent with what is known from SF in osteoarthritis.

Thus, among crystal-positive synovial samples, about two-thirds were associated with arthritis, with CPP arthritis being the most prevalent (about 60%). Extrapolation of these results to the overall synovial cohort suggests an overall prevalence of crystal arthritides in our cohort of 27%.

### 3.3. Patient Characteristics of Crystal Positive and Crystal Negative Samples

As can be seen in Table 3, men showed much higher prevalences for MSU than women (ratio up to 10.1:1 in inflammatory samples), whereas for CPP prevalence was very similar (up to 1.4:1). Numbers for ARP crystals were almost equilibrated, but numbers were very low. Of note, mean ages and age ranges were comparable between inflammatory and non-inflammatory samples in men (mean 62.4, range 18–94 years vs. mean 61.7, range 18–95 years) and women (mean 60.7, range 18–94 years vs. 65.1, range 21–93 years), respectively.

In line with the current CPPD classification criteria, the fraction of patients with CPP arthritis > 60 years of age was markedly higher than in all other crystal-positive cohorts (80.4% vs. 56–72.5%). This pattern suggests that biological factors associated with male sex and higher age are involved in the development of CPP arthritis.

Notably, however, 19.6% of inflammatory CPP-positive samples were derived from patients < 60 years of age, suggesting that CPP arthritis still should be considered in patients not matching the EULAR criteria.

### 3.4. Cellular Characteristics in Crystal-Positive Synovial Fluids

Next, we analyzed the fraction of Polymorphonuclear cells (PMN %) in inflammatory and non-inflammatory synovial fluid samples (Figure 2).

As expected, median PMN% were significantly higher in all inflammatory samples, when compared to non-inflammatory samples. However, even in non-inflammatory MSU-positive samples, PMN % yields were significantly higher than non-inflammatory ARP-, CPP-containing as well as in crystal-negative (i.e., examined but none detected) samples. This suggests subclinical immune stimulation by MSU, but not CPP and ARP crystals, without fulfilling the formal criteria for synovitis.

### 3.5. Localization of Crystals

Finally, we also compared whether qualitative (intracellular, extra/intracellular, extracellular) descriptions by observers regarding crystal localization correlated with the state of inflammation (Appendix A).

Of note, no quantification tools were used, and the description barely relied on the observer’s discretion. Nonetheless, in both MSU- and CPP-positive samples, we found a trend towards higher yields of intracellular crystals in inflammatory samples, suggesting increased phagocytosis and cellular stimulation. However, yields of intracellular and intra-/extracellular crystal localization were also high in formally non-inflammatory samples (72–77%). Because the assessment of crystal localization is highly operator-dependent and only semi-quantitative at the most, we did not perform statistical analysis.

## 4. Discussion

Crystal arthritides lead to significant suffering for the affected patients and to extensive socioeconomic disease burden in Western societies [17,18]. The prevalence of crystal arthritides is on the rise, and the relevance of these diseases for cardiovascular morbidity and mortality is increasingly acknowledged [13,14,15,16]. Nevertheless, studies on the prevalence of the different forms of crystal arthritis are scarce or focused on one crystal type. To our knowledge, we herein provide the most comprehensive side-by-side comparison of MSU-, CPP- and ARP SF samples with regard to originating joints, patient characteristics, and synovial cellular responses in one of the largest reported cohorts. In this clinically unbiased retrospective analysis, we found an overall prevalence of 39.3% of crystal-positive SF samples, with a 27% prevalence of crystal arthritis. Among crystal-positive samples, the highest prevalence was found for CPP crystals, accounting for 58.9% of inflammatory and 65.7% of non-inflammatory samples, respectively. Of note, MSU crystals were about twice more common in inflammatory than in non-inflammatory samples (43.4% vs. 23.4%).

There are only a few studies that are comparable in size. In fact, the oldest derives from our institution by Schlapbach et al. pursuing a comparable approach in 4475 samples [19]. Although they also found more CPP- than MSU-positive SF samples, prevalences were only 13.2% and 10.9%, respectively. This is remarkable as they investigated markedly more aspirates from knee joints than we did (82.9% vs. 48.7%), even if one considers that approximately 50% of “other” structures in our study likely correspond to knee joints. Oliviero et al. studied 2370 samples based on previous rheumatological diagnoses [12]. Among the 35% of non-inflammatory SF samples from osteoarthritis patients, 22.3% and 0% were CPP- and MSU-positive, respectively. This was in stark contrast to 65.7% CPP and 23.4% MSU positivity of non-inflammatory samples in our study, while patient characteristics in non-inflammatory and inflammatory cohorts seemed similar in both studies. Lastly, the study by Heselden et al. included 6983 samples [20], of which 52.8% and 40.8% were positive for CPP and MSU, respectively, thus matching our results. Taken together, the prevalences of CPP crystals in comparable studies are highly variable, but CPP prevalences in our study seemed to be the highest, and in line with high CPP prevalences detected by ultrasound [21]. Also, in our study, 6.1% of samples were MSU/CPP double positive, compared to the previously reported 2.5% [18]. These differences may partly be explained by a possible bias in underlying diagnoses and indications leading to arthrocentesis among the different studies. The predominance of male patients (up to 10.1:1) among MSU arthritis samples and a clear age shift in patients with CPP arthritis in our study was in line with previous studies, suggesting that our cohort was not undergoing a general selection bias. Conversely, missing information on the investigated joints in some of the above-mentioned studies may also contribute to a bias in CPP prevalences. Specifically, 52.5% of all samples in our study were derived from knee and wrist joints, both of which represent the most affected anatomical sites for chondrocalcinosis [4,17].

We found a clear shift towards higher ages in inflammatory CPP-positive samples when compared to non-inflammatory CPP-positive samples. Interestingly, this shift was only seen in men. This suggests that there are age- and sex-dependent factors that mediate the transformation from chondrocalcinosis to CPP arthritis. Whether this could be explained by higher levels of priming or activity of the NLRP3 inflammasome in the elderly [22,23], particularly in men, needs to be explored in future studies.

MSU-positive samples were significantly more often associated with an inflammatory cell pattern. While this is in line with the high immunogenic potential of MSU crystals, a more pronounced clinical presentation may have elevated the likelihood of arthrocentesis in these patients. Vice versa, the absence of clinical symptoms may have lowered the probability of detecting intercritical gout and, thus, may have shifted our results towards an elevated association between the presence of MSU crystals and an inflammatory state.

While current ACR/EULAR recommendations primarily recommend the treatment of established gout arthritis [24,25], the treatment of asymptomatic hyperuricemia remains controversial. Smaller studies from Andrés et al. and Pascual et al. showed elevated leukocytes in SF in preclinical [26] and intercritical [27] gout. Moreover, it has been previously shown that proinflammatory cytokines are elevated in asymptomatic hyperuricemia [28] when MSU deposits are already present. In our study, the presence of MSU crystals resulted in a higher fraction of inflammatory samples and PMN yields than CPP, ARP, and crystal-negative samples. This supports a model in which asymptomatic crystal deposition leads to subclinical inflammation, possibly contributing to atherosclerosis and cardiovascular morbidity. Thus, the presence or absence of clinical arthritis alone may not be adequate to assess elevated cardiovascular risk. New peripheral or intraarticular biomarkers may, therefore, be necessary for treatment decisions in the future. In fact, the recently proposed “treat-to-dissolve” instead of the common treat-to-target (based on uric acid levels in serum) strategy in gout [29] may be more adequate to treat high-risk patients.

Of note, our data on elevated PMN fractions in non-inflammatory, CPP-positive samples pointed in a similar direction and in line with previous studies [30]. However, significance when comparing to ARP or control samples was not reached. Again, this is in line with previous studies indicating higher PMN yields [12] and elevated proinflammatory cytokines (IL-1beta, IL-1RA, IL-8) [31] in CPP-positive SF samples from OA patients. Several explanations exist as to why PMN fractions in CPP-positive non-inflammatory samples were less pronounced than in the MSU-positive counterparts. MSU crystals show a higher inflammatory potential when compared to t-CPP, a-CPP, and m-CPPTβ phases [32]. By contrast, the monoclinic (m-)CPP phase induces IL-1 secretion comparable to MSU. Thus, both the yields of PMN as well as the development of frank synovitis by CPP directly depend on the composition of the different CPP phases and possibly other proinflammatory crystal forms [33,34]. In analogy to gout, we, therefore, hypothesize that there are preclinical and intercritical states in pseudogout, which may be relevant contributors to cardiovascular comorbidities.

Our study has limitations because of its retrospective design and the missing information on primary diagnoses, comorbidities, treatments, as well as clinical symptoms leading to arthrocentesis. Thus, our results may not apply to non-symptomatic or non-clinically effusive joints. Although we pursued an unbiased study, this approach bears the risk of over- or underrepresentation of certain patient groups. Also, currently, we cannot exclude a certain bias by repeating samples from the same patient at different timepoints. Concomitant septic arthritis may be present in a few cases but is unlikely to have a major impact on the overall prevalence of the different crystal types. The prevalence of septic arthritis in our cohort is currently under investigation. Finally, in a central laboratory setting, such as ours, minimal amounts of SF (approximately 1cc) are required to perform cellular and crystal analysis side by side. Thus, a relative underrepresentation of small joints (wrist, fingers/toes) leads to a possible bias when compared to big joints (e.g., knee). As knee joints are easier to puncture and yield larger amounts of SF, they also allow more comprehensive SF analysis. Also, as ARP detection requires an additional staining step with alizarin-red and sample volume, this exam is often not requested. This may contribute to the very low ARP sample numbers in our cohort, which makes conclusions on prevalence unreliable. We assume that these limitations are true for most studies comparable to ours. However, in our study, significantly more ARP was non-inflammatory, which would be consistent with osteoarthritis of the affected joint, while inflammatory samples may have been derived from patients with Milwaukee shoulder syndrome. This remains speculative, as we could not analyze the patients underlying diagnoses.

The strengths of our study include the large sample size from a real-world scenario, side-by-side comparison of the three most relevant crystal types, as well as distribution of affected joints. In addition, the fact that epidemiological data on sex and age distribution were comparable to most previous studies may imply that our monocentric study may indeed have relevance for other centers.

Taken together, our data suggest that the prevalences of CPP crystals in synovial fluids and CPP arthritis are high and possibly higher than generally assumed. Consequently, CPP arthritis should commonly be considered as a differential diagnosis in elderly patients with unclear inflammation. In addition, our data support the notion that CPP arthritis may be a common confounder in elderly patients with another underlying rheumatological disease, e.g., (seronegative) rheumatoid arthritis [10,11,35]. Further studies are necessary to better understand the clinical relevance of subclinical inflammation in MSU and CPP deposition and should include a definition of additional biomarkers.

## Acronyms

SFsynovial fluid samplesMSUMonosodium UrateCPPCalciumpyrophosphateARPalizarin-red positivePMNPolymorphonuclear cellsCCchondrocalcinosisCRconventional radiography

## Figures and Tables

**Figure 1 diagnostics-15-00907-f001:**
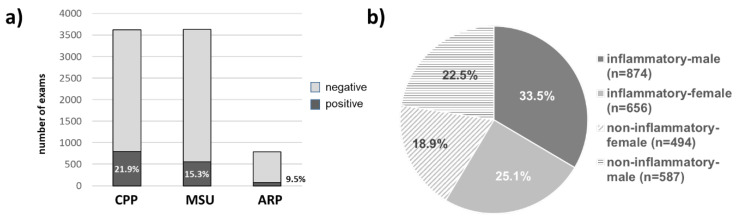
Basic characteristics of analyzed synovial fluid (SF) samples (*n =* 4295). (**a**) Prevalence of crystal-positive samples among all analyzed samples; (**b**) distribution of inflammatory and non-inflammatory according to sex among samples with available cellular analysis (*n =* 2611). CPP, calcium-pyrophosphate; MSU, monosodium urate; ARP, Alizarin-red positive.

**Figure 2 diagnostics-15-00907-f002:**
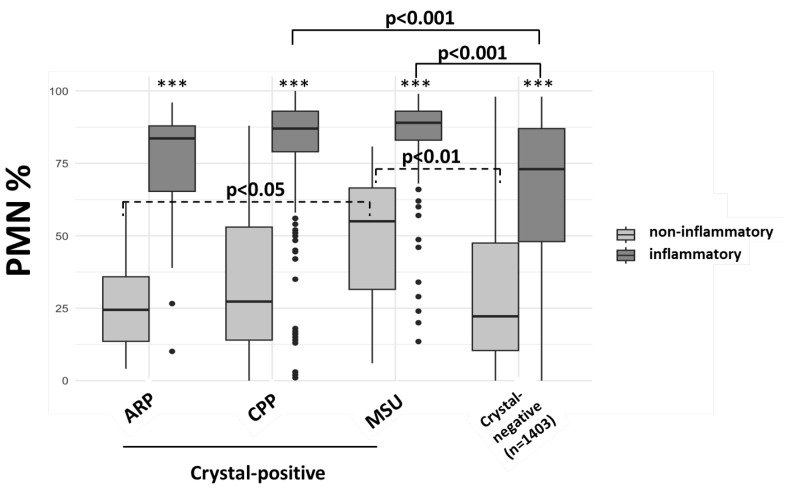
Polymorphonuclear (PMN) fractions in non-inflammatory and inflammatory samples, either containing MSU, CPP, or ARP crystals or crystal-negative samples. *** *p* < 0.0001 by Kruskal–Wallis test.

**Table 1 diagnostics-15-00907-t001:** Prevalence of crystal types according to joint origin (*n =* 1325).

	MSU (*n =* 471)	MSU/CPP (*n =* 80)	CPP (*n =* 699)	ARP/CPP (*n =* 13)	ARP (*n =* 58)
Shoulder	4 (0.8%)	0	38 (5.4%)	3 (23.1%)	15 (25.9%)
Wrist	25 (5.3%)	7 (8.9%)	39 (5.6%)	0	4 (6.9%)
Hip	4 (0.8%)	0	18 (2.6%)	0	6 (10.3%)
Knee	196 (41.6%)	47 (58.8%)	359 (51.4%)	9 (69.2%)	19 (32.8%)
Ankle	44 (9.3%)	4 (5.1%)	14 (2.0%)	0	0
MTP I	21 (4.5%)	1 (1.2%)	2 (0.3%)	1 (7.7%)	1 (1.7%)
Fingers/toes	27 (5.7%)	2 (2.4%)	13 (1.9%)	0	1 (1.7%)
Bursae	11 (2.3%)	1 (1.2%)	11 (1.6%)	0	3 (5.2%)
Other	139 (29.5%)	18 (22.5%)	205 (29.3%)	0	9 (15.5%)
**Prevalence among crystal-positive** ^†^	**35.7%**	**6.1%**	**52.9%**	**1%**	**4.4%**

^†^ Not considered MSU/ARP *n =* 2, MSU/ARP/CPP *n =* 2.

**Table 2 diagnostics-15-00907-t002:** Distribution of crystals or crystal constellations among non-inflammatory and inflammatory crystal-positive samples (*n =* 641); *p*-values indicate differences in crystal prevalences between non-inflammatory and inflammatory samples, respectively. Percentages indicate the relative prevalence of each crystal type or constellation among non-inflammatory or inflammatory samples, respectively.

	MSU	MSU/CPP	CPP	CPP/ARP	ARP	MSU/ARP	MSU/CPP/ ARP
Non-inflammatory (*n =* 201)	41 (20.3%)	5 (2.5%)	120 (59.7%)	7 (3.4%)	26 (12.9%	1 (0.5%)	1 (0.5%)
Inflammatory (*n =* 440)	162 (36.8%)	29 (6.6%)	224 (50.9%)	6 (1.4%)	18 (4.1%)	0 (0%)	1 (0.5%)
*p*-value	<0.001	ns	ns	ns	<0.001	ns	ns

ns, non-significant; CPP, calcium-pyrophosphate; MSU, monosodium urate; ARP, Alizarin-red positive.

**Table 3 diagnostics-15-00907-t003:** Epidemiological data of patients with and without non-specific crystal arthritides vs. crystal-positive non-inflammatory and crystal-negative samples, only mono-specific crystal-positive samples were considered.

	Non-Inflammatory				Inflammatory			
	Crystal Neg ^†^ (*n =* 646)	MSU (*n =* 41)	CPP (*n =* 120)	ARP (*n =* 25)	Crystal Neg ^†^ (*n =* 800)	MSU (*n =* 162)	CPP (*n =* 225)	ARP (*n =* 18)
male	53%	84%	48%	26%	48%	91%	59%	64%
Mean age male (yrs)	61.5	60.8	66.2	63.4	58.2	62.2	71.7	62.6
Mean age female (yrs)		70.8	71.5	74.1		71.7	70.8	70.2
>60 yrs	371 (57.6%)	23 (56.0%)	87 (72.5%)	18 (72.0%)	408 (51%)	93 (57.4%)	181 (80.4%)	11 (61.1%)
50–60 yrs	167 (25.9%)	9 (22.0%)	27 (22.5%)	5 (20.0%)	178 (22.2%)	37 (22.8%)	26 (11.6%)	3 (16.7%)
<50 yrs	108 (16.8%)	9 (22.0%)	6 (5.0%)	2 (8.0%)	214 (26.8%)	32 (19.8%)	18 (8.0%)	4 (22.2%)

^†^ ARP not detected or no alizarin-red staining performed. CPP, calcium-pyrophosphate; MSU, monosodium urate; ARP, Alizarin-red positive.

## Data Availability

The data presented in this study are available on request from the corresponding author.

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
