# Peer review of "Predominance of Calcium Pyrophosphate Crystals in Synovial Fluid Samples of Patients at a Large Tertiary Center"

_diagnostics, 2025, doi:10.3390/diagnostics15070907_

Round 1

Reviewer 1 Report

Comments and Suggestions for Authors

1
I read with interest this paper about a large series of synovial fluid samples with a focus on crystal prevalence. I have the following comments for the authors:
Line 51: Milwaukee shoulder is predominantly low symptomatic, with mechanical pain and large effusions. Please correct that.
Line 58-59: There are recent works proving DECT can distinguish between calcium containing crystals [see by instance doi:10.1016/j.joca.2019.05.007], so the statement included by the authors is not accurate. Please amend it.
Line 132 (Figure 1): I find this Figure really confusing. Sankey plots are usually used for data changing over time, but here is pretended for stratifications, but the result is not satisfactory from my perspective and may be misinterpreted by the reader. I suggest a flow diagram, initiating from SF samples and the distribution per groups of crystals seen; then, indicate % of inflammatory fluids, gender, age (please include, perhaps as % >60yo) and location.
Added to it, these data represent Results and must be replaced accordingly.
Line 147: Besides gender, please give also here the average age of the patients
Line 173: Explain in Table 1 footnotes the meaning for the grey shadowed cells and, if not mandatory, please remove.
Line 183: Similarly, in Table 2, remove text in bold for CPP crystals if not mandatory.
Line 188 and 211: Sections 3.2 and 3.3 are too long and most of the data given is already depicted in the accompanying Tables 2 and 3. I recommend the authors to shorten them.
Line 218 (Table 3): Please add a dispersion measure for age. Here, instead of SD, I think range would be more informative to better depiction of the population.
Line 305: For this paragraph, the authors should consider whether in the center there might be the practice to avoid joint aspiration to diagnose gout out of flares (intercritical). That also might explain the more prominent association with inflammatory fluids.
Line 323: Please, include for this discussion the following work by Martinez-Sanchis and Eliseo (doi: 10.1136/ard.2005.035386).
Please amend some typos in lines 124 (“polymorphnu-“), 192 (“thant”), 195 (“thritdes”), 451 (“und”)

Author Response

I read with interest this paper about a large series of synovial fluid samples with a focus on crystal prevalence. I have the following comments for the authors:
Line 51: Milwaukee shoulder is predominantly low symptomatic, with mechanical pain and large effusions. Please correct that.

Thank you for the comment, we have corrected the sentence.

Line 58-59: There are recent works proving DECT can distinguish between calcium containing crystals [see by instance doi:10.1016/j.joca.2019.05.007], so the statement included by the authors is not accurate. Please amend it.

Thank you, we have changed the sentence and added two references

Line 132 (Figure 1): I find this Figure really confusing. Sankey plots are usually used for data changing over time, but here is pretended for stratifications, but the result is not satisfactory from my perspective and may be misinterpreted by the reader. I suggest a flow diagram, initiating from SF samples and the distribution per groups of crystals seen; then, indicate % of inflammatory fluids, gender, age (please include, perhaps as % >60yo) and location. Added to it, these data represent Results and must be replaced accordingly.

We thank the reviewer for the comments and have placed the figure in the Supplementary material. However, in our opinion the Sankey blot nicely indicates the sex distribution and, more importantly, the originating joints in our SF cohort. In our opinion the information suggested by the reviewer is better shown in Table 1 (joint origin per crystal type), Table 2 (distribution of crystals and % of inflammatory/non-inflammatory) as well as in Table 3 (age distribution, gender, age >60years per crystal type and per inflammatory/non-infalmmatory). Depicting all this in a new Sankey blot would be rather more complex.

Line 147: Besides gender, please give also here the average age of the patients

We have now added the requested details regarding age distribution

Line 173: Explain in Table 1 footnotes the meaning for the grey shadowed cells and, if not mandatory, please remove.

We referred to the grey cells in the text, but do not think it is mandatory. We removed the shadowing and description in the text, accordingly.

Line 183: Similarly, in Table 2, remove text in bold for CPP crystals if not mandatory.

We corrected the table, as suggested

Line 188 and 211: Sections 3.2 and 3.3 are too long and most of the data given is already depicted in the accompanying Tables 2 and 3. I recommend the authors to shorten them.

We shortened the section, as suggested.

Line 218 (Table 3): Please add a dispersion measure for age. Here, instead of SD, I think range would be more informative to better depiction of the population.

We have added a sentence regarding mean age and age distributionin men and women and in inflammatory and non-inflammatory samples.

Line 305: For this paragraph, the authors should consider whether in the center there might be the practice to avoid joint aspiration to diagnose gout out of flares (intercritical). That also might explain the more prominent association with inflammatory fluids.

We thank the reviewer for this comment. Although we cannot retrospectively assess the decision to not perform arthrocentesis, we included this important point in the discussion.

Line 323: Please, include for this discussion the following work by Martinez-Sanchis and Eliseo (doi: 10.1136/ard.2005.035386).

Thank you for this reference, which we included.

Please amend some typos in lines 124 (“polymorphnu-“), 192 (“thant”), 195 (“thritdes”), 451 (“und”)

Typos were corrected. Thank you.

Reviewer 2 Report

Comments and Suggestions for Authors

diagnostics-3526147-peer-review

Calcium Pyrophosphate Crystals in Synovial 2 Fluid

General comments: a very large sample study.

1. Page 1, sentence 1 “Crystal arthritides by monosodium urate (MSU-arthritis, “gout”) or calciumpyrophosphate (CPP-arthritis, “pseudogout”) crystal deposits represent the most prevalent causes for arthritis.”  - please change this to “inflammatory arthritis”.

2. Similarly next sentence: “gouty arthritis is the most common arthritis form” change to “inflammatory arthritis form”. Osteoarthritis is the most common arthritis form.

3. Why is the abbreviation for “basic calcium phosphate” - ARP? Why not BCP?

4. Since these were clinical samples, it can be assumed that these arthrocenteses were performed because of a symptomatic and usually a suspected effusive joint.   Thus, these results would not apply to non-symptomatic or non-clinically effusive joints. Please note this in the limitations.

5. These data also indicate that crystal examination is critical, as fluids from symptomatic joints have such a high yield for these crystals. Often orthopedic surgeons are primarily focusing on excluding infection, rather than excluding crystal disease - these results suggest that crystal examination is absolutely necessary even in suspected septic arthritis.

6. Did the authors exclude septic fluids from these analyses?  Septic fluids can also have crystals, and preexisting crystal arthritis appears to be a risk factor for septic arthritis. Please clarify the question of septic fluids and crystals in the both methods and limitations.

Author Response

  1. Page 1, sentence 1 “Crystal arthritides by monosodium urate (MSU-arthritis, “gout”) or calciumpyrophosphate (CPP-arthritis, “pseudogout”) crystal deposits represent the most prevalent causes for arthritis.”- please change this to “inflammatory arthritis”.
  2. Similarly next sentence: “gouty arthritis is the most common arthritis form” change to “inflammatory arthritis form”. Osteoarthritis is the most common arthritis form.

We appreciate the comments, but think there is a mismatch between European and American terminology. In Europe the latin term “arthritis” is per se defined as inflammatory. Thus, “inflammatory arthritis” would double express this characteristic. In the same context the term “osteoarthritis”, which is per definition non-inflammatory, does not make much sense from a European perspective. As we are a European centre, we would like to stick to the European nomenclature.

  1. Why is the abbreviation for “basic calcium phosphate” - ARP? Why not BCP?

Thank you for the comment. We changed this from BCP to ARP after several reviewers have criticized the term BCP as being not equivalent to Alizarin-red positive crystals, since CPP crystals are also staining positive for alizarin-red.

  1. Since these were clinical samples, it can be assumed that these arthrocenteses were performed because of a symptomatic and usually a suspected effusive joint. Thus, these results would not apply to non-symptomatic or non-clinically effusive joints. Please note this in the limitations.

Thank you for this comment. We have included this in our Discussion (L325-327 in the revised version).

  1. These data also indicate that crystal examination is critical, as fluids from symptomatic joints have such a high yield for these crystals. Often orthopedic surgeons are primarily focusing on excluding infection, rather than excluding crystal disease - these results suggest that crystal examination is absolutely necessary even in suspected septic arthritis.

We absolutely agree and have added an additional sentence in the discussion to emphasize this point.

  1. Did the authors exclude septic fluids from these analyses?Septic fluids can also have crystals, and preexisting crystal arthritis appears to be a risk factor for septic arthritis. Please clarify the question of septic fluids and crystals in the both methods and limitations.

The reviewer raises an important point, which is currently being investigated in our SF cohort. However, from literature it should not be expected that more than 1% of crystal-positive patients actually is simultaneously septic. We have adapted the discussion, accordingly (L362-365 in the revised version).

Reviewer 3 Report

Comments and Suggestions for Authors

I appreciated the opportunity to review this retrospective study aimed to assess the prevalence and characteristics of crystal arthritides at the largest tertiary center in Switzerland. 

I just have some comments that I think can help improve the manuscript overall:

  • The authors should define better how a sample is declared  inflammatory vs. non-inflammatory samples (it could be useful to state specific cell count thresholds).
  • Was the microscopic analysis conducted blinded to clinical information to minimize the risk of bias?

Author Response

  • The authors should define better how a sample is declared  inflammatory vs. non-inflammatory samples (it could be useful to state specific cell count thresholds).

We had the definition already included in the methods section, see lines 127-128

  • Was the microscopic analysis conducted blinded to clinical information to minimize the risk of bias?

Thank you for this comment. Yes, in most cases the examintors were blinded to clinical information. As some of the samples were analyzed by rheumatologists, who were possibly involved in managing the cases, we cannot exclude this for 100% of the cases.  This is, however, impossible to find out in our retrospective cohort analysis.

Round 2

Reviewer 1 Report

Comments and Suggestions for Authors

I have no further comments; the manuscript has improved significantly.

Reviewer 2 Report

Comments and Suggestions for Authors

I believe the authors did a good job revising the manuscript.